# Effect of Core–Shell Rubber Nanoparticles on the Mechanical Properties of Epoxy and Epoxy-Based CFRP

**DOI:** 10.3390/ma15217502

**Published:** 2022-10-26

**Authors:** Tatjana Glaskova-Kuzmina, Leons Stankevics, Sergejs Tarasovs, Jevgenijs Sevcenko, Vladimir Špaček, Anatolijs Sarakovskis, Aleksejs Zolotarjovs, Krishjanis Shmits, Andrey Aniskevich

**Affiliations:** 1Institute for Mechanics of Materials, University of Latvia, Jelgavas 3, LV-1004 Riga, Latvia; 2Synpo, S. K. Neumanna 1316, 530 02 Pardubice, Czech Republic; 3Institute of Solid State Physics, Kengaraga 8, LV-1063 Riga, Latvia

**Keywords:** epoxy, CFRP, core–shell rubber nanoparticles, tensile properties, fracture toughness, glass transition temperature

## Abstract

The aim of the research was to estimate the effect of core–shell rubber (CSR) nanoparticles on the tensile properties, fracture toughness, and glass transition temperature of the epoxy and epoxy-based carbon fiber reinforced polymer (CFRP). Three additives containing CSR nanoparticles were used for the research resulting in a filler fraction of 2–6 wt.% in the epoxy resin. It was experimentally confirmed that the effect of the CSR nanoparticles on the tensile properties of the epoxy resin was notable, leading to a reduction of 10–20% in the tensile strength and elastic modulus and an increase of 60–108% in the fracture toughness for the highest filler fraction. The interlaminar fracture toughness of CFRP was maximally improved by 53% for ACE MX 960 at CSR content 4 wt.%. The glass transition temperature of the epoxy was gradually improved by 10–20 °C with the increase of CSR nanoparticles for all of the additives. A combination of rigid and soft particles could simultaneously enhance both the tensile properties and the fracture toughness, which cannot be achieved by the single-phase particles independently.

## 1. Introduction

Epoxy resins having relatively high tensile strength and modulus of elasticity, a low creep, and a good stability at elevated temperatures are extensively used as matrices in composite technology for different applications [1,2]. Nevertheless, due to high crosslinking, they are characterized by a high degree of brittleness and a poor resistance to crack initiation/propagation [3].

Their toughness could be improved by adding core–shell rubber (CSR) nanoparticles that are made of a soft rubbery core and a rigid shell around it which are mainly manufactured by emulsion polymerization and then added to the polymer resins. In comparison with the phase-separating rubbers, this method allows the advantage of controlling the particle size by changing the core and shell diameters [4]. The materials that are usually used for the core are siloxane, butadiene, and acrylate polyurethane, while poly (methyl methacrylate) (PMMA) is preferred to be used as the shell materials due to it having a good compatibility with the epoxy polymers [5,6].

It was determined that the addition of CSR particles led to a significant reduction in the tensile properties of the epoxy resin (DGEBA) and almost no effect on its glass transition temperature (*T*_g_) [3]. For the 15 wt.% content of CSR in the epoxy, the elastic modulus and tensile strength of the epoxy were diminished by 27 and 36%, respectively. However, for the same composition of CSR filler particles, the fracture energy was improved by 550%. Similar results were obtained for the epoxy that was filled with CSR particles from 0 to 38 vol.%, revealing a gradual increase in the *T*_g_ and Poisson’s ratio and a significant decrease in the tensile and compressive properties of the CSR-modified composites which were explained by rubber having a lower Young’s modulus and a higher Poisson’s ratio in comparison with the epoxy [7]. By using SEM of fracture surfaces and analytical models, several toughening mechanisms (shear band yielding, core-to-shell debonding and plastic void growth) were defined [3,7].

In general, the fracture toughness of epoxy was improved by adding both rigid and soft particles [8,9]. The rigid particles toughen the epoxy through crack pinning and crack deflection/bifurcation effects, while the toughening mechanisms of the soft particles are filler-debonding, and the subsequent void grows as well as the matrix shear band.

The research aimed to estimate the effect of core–shell rubber (CSR) particles on the tensile properties, fracture toughness and glass transition temperature of the epoxy and epoxy-based CFRP. The novelty of this work is in the multi-step approach for the evaluation of the toughening effects for both the epoxy and epoxy-based CFRP and considering their mechanical properties. The application of the proposed solution with improved fracture toughness both for the epoxy and epoxy-based CFRP could broaden their use in aerospace, automotive, marine and sporting goods due to them having a longer lifetime and enhanced safety features.

## 2. Materials and Methods

### 2.1. Materials

CHS-Epoxy 582 (Spolchemie, Usti nad Labem, Czech Republic) [10] was used as matrix material. It is a diglycidyl ether of bisphenol A (DGEBA) with a reactive diluent that has an epoxide equivalent weight (EEW) of 165–173 g/mol. This epoxy resin is recommended for different applications in composites, adhesives, wind energy, construction, electronics and corrosive coatings. The hardener Telalit 0420 (Spolchemie, Usti nad Labem, Czech Republic) which is a cycloaliphatic amine was mixed with epoxy resin at a ratio of 100:25 [11].

Three additives containing CSR nanoparticles which were dispersed in DGEBA with different particle sizes and core material ACE MX 125, 156 and 960 were supplied by Kaneka (Westerlo, Belgium). The information regarding core material and CSR size are given in Table 1. For all of the additives that were studied, the concentration of CSR nanoparticles in DGEBA was 25 wt.%, the shell material was PMMA, and the density was 1.1 g/cm^3^ [12]. Carbon fiber fabric KC (0/90) in plane weave and of a specific surface of 160 g/m^2^ was supplied by Havel Composites (Svésedlice, Czech Republic) [13] and used for the manufacturing of CFRP laminates.

### 2.2. Manufacturing of the Test Samples

For pure epoxy samples, the epoxy resin was manually mixed with the hardener for approx. 10 min and the mixture was further degassed by using the vacuum pump. For CSR-modified epoxy resin, a certain content of CSR nanoparticles (2, 4, and 6 wt.%) was added to the epoxy and manually mixed, degassed, and then mixed with the hardener for approx. 10 min. After degassing, all of the mixtures were poured into silicon molds. The curing and post-curing conditions were chosen based on supplier recommendations [10]: overnight at room temperature (RT), 2 h at 60 °C, 1 h at 80 °C, and 1 h at 120 °C.

The silicon molds were used for the manufacturing of the test samples to determine the tensile properties [14,15] and fracture toughness [16] of the epoxy and epoxy modified with CSR particles. Thus, five dog-bone samples and five tapered double cantilever beam (TDCB) samples were manufactured for each test and CSR particle type and each filler fraction.

Double cantilever beam (DCB) CFRP samples were produced by lay-up technology by using woven carbon fiber fabric (0/90)_12_, which was cured at RT, cut into samples and post-cured as CSR-modified epoxy resin. The CSR nanoparticle fraction of 4 wt.% in the epoxy resin was used for the manufacturing of all of the CFRP plates based on the highest results of fracture toughness obtained for the modified epoxy in TDCB tests. At least five DCB samples were manufactured and tested for each CSR nanoparticle additive.

### 2.3. Testing Methods

#### 2.3.1. Morphology Analysis

The morphology of the fracture surfaces for CFRP samples was examined by using a high-resolution SEM-FIB electron microscope Helios 5 UX (Thermo Scientific, Walthamm, MA, USA), which was operated at 1 kV and 25 pA with scan interlacing and integration to avoid charging.

#### 2.3.2. Tensile Tests

For the test specimens of epoxy and epoxy that was modified with CSR nanoparticles, quasi-static tensile tests were performed by using Zwick 2.5 universal testing machine with a crosshead speed of 2 mm/min at RT. The tensile strength was defined as the maximal achieved value of stress in the specimen, and the elastic modulus was calculated from the slope of a secant line between 0.05 and 0.25% strain on a stress–strain plot. Five test samples per each CSR type and fraction were tested, and the values that are provided correspond to their arithmetic mean value.

#### 2.3.3. Fracture Toughness Tests

A specimen with a sharp pre-crack is needed for the precise measurement of the stress intensity factor (SIF). TDCB specimens produced in the silicone molds had an initial notch with a 1 mm width and a round end, which may substantially increase the apparent fracture toughness of the material. Therefore, the initial pre-crack of 2–5 mm length was made in the specimen before testing by the sharp knife strike. Moreover, side grooves of a depth of approx. 2 mm were produced to minimize the crack deflection and to keep the crack path along the midplane of the specimens [16]. The tests were conducted on Zwick 2.5 universal testing machine at RT with a constant displacement rate of 1 mm/min. SIF was calculated using Mode I load for a crack length < 20 mm within a constant SIF region.

For the specimen without side grooves, the SIF can be evaluated as follows [16]:(1)$$K_{ng}=2P_{c}\frac{\sqrt{m}}{b},$$
where *P*_c_ is the critical load, *b* is the width of the specimen, and *m* is a geometrical parameter, which for the specimen of the considered geometry equals 0.6 mm^−1^. For the specimen with side grooves, Equation (2) should be modified as
(2)$$K_{g}=K_{ng}{\left(\frac{b}{b_{n}}\right)}^{0.56},$$
where *b**_n_* is the reduced width of the specimen at the grooves’ location, and the exponent value was determined from a series of 3D finite element simulations with grooves of different depths (see Appendix A).

#### 2.3.4. Interlaminar Fracture Toughness Tests

The Mode I interlaminar fracture toughness tests of carbon 0/90 woven fabric laminates were carried out according to ASTM: D5528 [17] using specimens with dimensions of 25 × 3 × 125 mm^3^. Though this standard was specified for unidirectional laminates, it has been successfully applied for laminates with different lay-up configurations [18]. According to this standard, a linear elastic behavior is assumed in the calculation of strain energy release rate, which is reasonable when the zone of damage at the delamination front is small relative to the thickness of the DCB sample. Opening Mode I interlaminar fracture toughness, *G*_IC_, was evaluated from the load–deflection curve at the point of deviation from linearity (NL). The NL calculation of *G*_IC_ considers that the delamination starts to grow from the insert in the interior of the specimen at this point. The tests were performed by using Zwick 2.5 testing machine with a crosshead speed of 1 mm/min at RT and Canon EOS40D to record photos every 3 s for the analysis of the crack propagation until a failure occurred. ImageJ 1.38x software [19] was used to estimate the delamination length in DCB samples. At least five DCB samples per each CSR type at 4 wt.% in the epoxy resin used for the impregnation of cross-ply CFRP laminates were tested.

The Modified Beam Theory [17] method was used for the calculation of Mode I interlaminar fracture toughness assuming the correction for the rotation at the delamination front (Δ):(3)$$G_{I}=\frac{3P\delta }{2b\left(a+\left|\Delta \right|\right)},$$
where *P* is the load, *δ* is the load point displacement, *a* is the delamination length, and Δ is determined experimentally by generating the least squares plot of the cube root of compliance as a function of delamination length.

Moreover, for the specimens with loading blocks, two correction parameters—a parameter *F* accounting for the shortening of the moment arm and the tilting of the end blocks and a displacement parameter *N* accounting for the stiffening of the specimen by the blocks—are recommended [17]:(4)$$F=1-\frac{3}{10}{\left(\frac{\delta }{a}\right)}^{2}-\frac{3}{2}\left(\frac{\delta t}{a^{2}}\right),$$
(5)$$N=1-{\left(\frac{L\prime }{a}\right)}^{3}-\frac{9}{8}\left[1-{\left(\frac{L\prime }{a}\right)}^{2}\right]\left(\frac{\delta t}{a^{2}}\right)-\frac{9}{35}{\left(\frac{\delta t}{a^{2}}\right)}^{2},$$
where *L′* and *t* are the geometrical parameters of the blocks.

Then, the corrected formula for interlaminar fracture toughness by using the Modified Beam Theory method takes the form:(6)$$G_{I}=\frac{3P\delta }{2b\left(a+\left|\Delta \right|\right)}\cdot \left(\frac{F}{N}\right).$$

#### 2.3.5. Density Measurements

The density of the epoxy and epoxy that was modified with CSR particles was defined at RT by using hydrostatic weighing in isopropyl alcohol and Mettler Toledo XS205DU balance with a precision of ±0.05 mg. First, the density of isopropyl alcohol was determined by using a sinker of a known volume of 10 cm^3^. Then, the mass of the samples was registered in the air (*m*_a_) and the liquid of known density (*m_l_*). The density of the samples was determined by the formula:(7)$$\rho =\frac{m_{a}}{m_{a}-m_{l}}\left(\rho_{l}-\rho_{a}\right)+\rho_{a},$$
where *ρ_l_* and *ρ_a_* are the densities of the liquid (0.785 g/cm^3^ for isopropyl alcohol) and air (0.0012 g/cm^3^), respectively.

#### 2.3.6. Thermal Mechanical Analysis

The glass transition temperature (*T*g) of the epoxy and epoxy modified by CSR particles was estimated by conducting thermomechanical analysis (TMA) tests using TMA/SDTA841e (Mettler Toledo, Greifensee, Switzerland). The samples were heated from 30 to 150 °C at a heating rate of 3 °C/min and a force of 0.02 N, and then, they were subsequently cooled. According to ASTM standard E1545 [20], the glass transition corresponds to the inflection in the dimensional change when plotted against the temperature upon which the material changes from a hard (brittle) state into a soft (rubbery) state. The glass transition temperature was evaluated as the extrapolated onset of the kink in the experimental TMA curve, which was displayed as a function of temperature. At least three tests were conducted for each CSR type and fraction, and the values that are provided correspond to their arithmetic mean value.

## 3. Results and Discussion

### 3.1. Morphology of the Fracture Surface

The microscopy analysis of the fracture surfaces of the pure epoxy-based CFRP shown in Figure 1a revealed smooth and glassy surfaces with straight and sharp crack paths, which are characteristic of a brittle damage property and a weak resistance to crack initiation and propagation [8]. No delamination on the interface between the carbon fibers and the epoxy resin was noticed. The fracture surfaces of all four wt.% CSR-modified compositions that are provided in Figure 1b–d proved that the dispersion of CSR nanoparticles was good, and no significant agglomeration of CSR nanoparticles was found. The diameter of the CSR nanoparticles which were evaluated using ImageJ software was slightly higher than the data that are provided in Table 1 by the manufacturer. For the additives ACE MX-125 and ACE-MX-156, the diameter was very similar, 126 ± 28 nm and 126 ± 26 nm, accordingly. In comparison with these two additives, the ACE MX-960 CSR particles were much larger and had a wide diameter scatter—440 ± 248 nm. It could be an indication that most of the CSR nanoparticles were debonded as particles’ debonding and subsequent plastic void growth is considered one of the most important toughening mechanisms for CSR/epoxy composites [3,8,9,21].

For all of the CSR nanoparticle additives (Figure 1b,c), it can be noticed that the fracture surface was much rougher, and the crack paths became more curved following the CSR-particle circular shape. It could be also observed that in comparison with the undamaged CSR nanoparticles, the ones on the crack path were not perfectly spherical, thereby revealing their valuable contribution to the crack propagation process [8].

### 3.2. Density and Porosity

The results that were obtained for the density of epoxy/CSR nanoparticle composites are shown in Figure 2. According to Figure 2, the addition of all of the additives containing CSR nanoparticles led to a decrease in the density of CSR-modified epoxy. By using the mixture rule, the density of the composite material could be estimated:(8)$$\rho_{c}=\rho_{f}\times v_{f}+\rho_{m}\times \left(1-v_{f}\right),$$
where *ρ_f_* and *ρ_m_* are the density of the filler (CSR nanoparticles) and polymer matrix (epoxy), respectively, and while *v_f_* is the volume fraction of the filler, accordingly. The density of the epoxy was experimentally found to be 1.159 ± 0.002 g/cm^3^. Considering the known density of the 25%-CSR-modified epoxy of 1.1 g/cm^3^ [12], the density of the CSR particles was found to be 0.91 g/cm^3^ [3]. Therefore, the addition of the filler particles of a lower density to the epoxy resin has resulted in a slight decrease (by approx. 2%) of the density for the composite. The higher the filler content was, then the lower that the density of the composite was.

The volume fraction of filler could be evaluated as follows [22]:(9)$$v_{f}=\frac{\rho_{m}\times c_{f}}{\rho_{m}\times c_{f}+\rho_{f}\times \left(1-c_{f}\right)},$$
where *c_f_* is the weight fraction of the filler.

As seen in Figure 2, by using the mixture rule (Equation (8)), an overestimated value for the density of all of the CSR-modified epoxy materials was obtained. Therefore, efforts were made to evaluate the density of the composites having additional phase, air-filled pores, which could exist in the composites, and as a result, could lead to them having a lower density:(10)$$\rho_{c}=\rho_{f}\times v_{f}+\rho_{p}\times v_{p}+\rho_{m}\times \left(1-v_{f}-v_{p}\right),$$
where *ρ_p_* is the density of the air and *v_p_* is the volume fraction of pores in the composites, respectively.

The volume fraction of the pores can be derived from Equation (10):(11)$$v_{p}=\frac{v_{f}\times \left(\rho_{f}-\rho_{m}\right)+\rho_{m}-\rho_{c}}{\rho_{m}-\rho_{p}}.$$

According to Figure 2, it is obvious that though the estimated volume fraction of the pores was only 0.8–2% by using the modified mixture rule (Equation (10)), a better correlation with the experimental data was obtained. It was used in the calculation of the elastic modulus of the epoxy that was filled with the CSR nanoparticles.

### 3.3. Tensile Properties and Glass Transition Temperature

The stress–strain curves for the epoxy and epoxy that was filled with the ACE MX-156 CSR particles are given in Figure 3a. Analogous results were obtained for the other additives. According to Figure 3b, the elastic modulus of all of the studied materials significantly decreased with the increasing CSR content. The elastic modulus of 1.99 ± 0.04 GPa was found for the unmodified epoxy. For the modified epoxy, it had the lowest value for ACE MX-960 at all of the filler fractions, which could be attributed to the lower effective stiffness of the particles due to the highest CSR size in comparison with the other additives (see Table 1) [2]. The tensile strength of the epoxy (73 ± 3 MPa) decreased by approx. 10–20% with the addition of the CSR particles. Again, slightly lower tensile strengths were found for ACE MX-960 in comparison to the other CSR nanoparticles. Moreover, it could be noted from Figure 3a that the maximal deformation increased (from 4.9 ± 0.6% to 7.2 ± 0.5%) with the increase of the CSR content, thereby revealing the plasticization/softening effect resulting from the inclusion of softer filler particles in a brittle matrix.

Several analytical models, e.g., the Halpin–Tsai [3], Lewis–Nielsen [3,7] and Mori–Tanaka ones [7], were used to predict the significant reduction of the elastic modulus for the epoxy that was filled with the CSR nanoparticles. Most models epitomize an ideal composite by making several assumptions, e.g., that the polymer matrix and the filler particles are linear-elastic and isotropic, thereby having a perfect bond between them [23,24,25]. Moreover, the porosity and agglomeration of the filler particles negatively affecting the mechanical properties are usually neglected. In this work, the Hansen model [26,27] considering the spherical particles that are embedded in spherical shells of the matrix was used. It was applied in two steps: 1. to estimate the elastic modulus of the epoxy matrix containing a certain volume fraction of the pores from Equation (8), and 2. to determine the elastic modulus of the epoxy (with the pores) that was filled with the CSR nanoparticles.

According to the Hansen model, the elastic modulus of the matrix that was filled with spherical particles was estimated by using the following formula:(12)$$E_{c}=\frac{\left(1-v_{f}\right)+\left(1+v_{f}\right)\;E_{f}/E_{m}}{\left(1+v_{f}\right)+\left(1-v_{f}\right)\;E_{f}/E_{m}}\times E_{m},\;$$
where *E_f_* and *E_m_* are the elastic moduli of the filler and the matrix, respectively.

For the first step considering the epoxy matrix that was filled with the pores (air bubbles), Equation (12) becomes simplified since $E_{f}/E_{m}\ll 1$, and it takes the form
(13)$$E_{c}^{I}=\frac{\left(1-v_{p}\right)}{\left(1+v_{p}\right)}\times E_{m}.\;$$

For the second step considering the epoxy matrix (with pores) that was filled with the CSR nanoparticles, Equation (12) was modified to include both pores and CSR nanoparticles
(14)$$E_{c}^{II}=\frac{\left(1-v_{f}\right)+\left(1+v_{f}\right)E_{f}/E_{c}^{I}}{\left(1+v_{f}\right)+\left(1-v_{f}\right)E_{f}/E_{c}^{I}}\times E_{c}^{I},\;$$
where the elastic modulus of the CSR particles *E_f_* = 4 MPa [2], and volume fraction of the filler and pores, *v_f_* and *v_p_*, were evaluated from Equations (9) and (11), respectively.

The results of the evaluation by Equations (12) and (14) are shown in Figure 3b. Generally, it could be concluded that at the higher filler contents, the Hansen model allowed us to predict the reduction of the elastic modulus by approx. 20% due to the addition of the soft CSR particles in the epoxy resin. It could be either noticed that the consideration of the pores (0.8–2.0 vol.%) improves the description of the experimental results. In general, the addition of ACE MX-960 to the epoxy resin led to marginally lower values of elastic modulus than those of the two other CSR-containing additives. It could indicate a higher volume of the softcore when it is compared to the total particle (core plus shell) volume since the size of these particles is the greatest when it is compared to the other ones (see Table 1).

The results that were obtained for the glass transition temperature as evaluated using the TMA diagrams are provided in Figure 4. The glass transition temperature of the epoxy was approx. 78.1 ± 2.2 °C which was within the range (70–140 °C) of the reported values of *T*_g_ for DGEBA type epoxy [3,8,28]. Contradictory results are provided in the literature revealing the occurrence of an improvement [7], a reduction [8] or almost no effect [3,4,21,28] on *T*_g_ for the epoxy with the addition of the CSR nanoparticles. According to Figure 4, a gradual increase of 10–20 °C was obtained for the epoxy that was filled with all of the additives containing the CSR nanoparticles, which could be attributed to the high crosslink density and toughening effect of rubber modifiers, thereby testifying to their dissolution in the epoxy continuous phase.

### 3.4. Fracture Toughness

The representative load–displacement curves for TDCB tests are provided in Figure 5a. Obviously, the soft CSR nanoparticles were effective as tougheners for the epoxy resin. According to Figure 5a, the critical load of the epoxy was significantly improved with the increase of CSR nanoparticles of ACE MX-960. Similar results were obtained for the rest of the additives containing the CSR nanoparticles.

The fracture toughness of the epoxy which was evaluated by Equation (2) was 0.83 ± 0.07 MPa·m^1/2^ which is slightly lower than the values that are reported in the literature for the epoxy resins [7,8]. As seen in Figure 5b, the addition of the CSR nanoparticles led to a gradual improvement in the fracture toughness for all of the types of additives. No considerable distinction in the fracture toughness among the additives was detected, thereby proving that small (100 nm) and large (300 nm) CSR particles were equally efficient. Though generally, ACE MX-156 showed the greatest enhancement in the fracture toughness value which was approx. 108% at the CSR content of 4 wt.%. The optimum rubber content beyond which the fracture toughness did not improve was reported [7,28]. In this work, according to Figure 5b, the optimum CSR nanoparticle content could be estimated as 4 wt.% for all of the additives. Of course, this result is only relevant for certain dispersion conditions of the CSR particles in the epoxy. Nevertheless, the manual mixing of the CSR nanoparticles in the epoxy resulted in a good dispersion of the CSR nanoparticles as seen by the SEM and a considerable improvement in the fracture toughness. Additionally, a low fraction of pores that was indirectly estimated from the density measurements revealed the sufficient quality of the manufactured samples.

### 3.5. Interlaminar Fracture Toughness

The typical DCB load vs. displacement curves of the unmodified and CSR-modified CFRP laminate specimens are shown in Figure 6a. The saw-like drops on the load–displacement diagrams after the critical load was achieved were obviously caused by the woven 0/90 lay-up configuration of the carbon fabric that was used to produce the composite laminate. According to Figure 6a, the effect of all of the additives containing the CSR nanoparticles was substantial, thereby leading to the improvement of the critical load of the CFRP by 32–70%. The Mode I interlaminar fracture toughness of CFRP which was evaluated by Equation (6) was enhanced from 390 ± 50 J/m^2^ to a maximal value of 599 ± 13 J/m^2^ as shown in Figure 6b for the CFRP with 4 wt.% of ACE MX-960.

However, the toughening effect of the CSR nanoparticles in the epoxy did not fully transfer to the epoxy-based CFRP composite laminates. E.g., the use of an epoxy that was modified with 4 wt.% of ACE-MX 156 having the maximal improvement of fracture toughness by 108% as a matrix for CFRP laminates resulted in the improvement to the interlaminar fracture toughness by only 32%. The interlaminar fracture toughness was maximally improved by 53% for ACE MX-960 at CSR content 4 wt.%. A further increase in the CSR fraction could result in greater improvement of the interlaminar fracture toughness of the CFRP, though, it should be emphasized that rubber toughening has also the side effect of increasing the viscosity of the epoxy resin, thereby negatively contributing to the fabrication of composite laminates [8,9,18,28]. Additionally, at higher values of the filler fraction, a significant agglomeration can occur, thereby causing a local stress concentration and a detrimental effect on the toughening performance of the filler particles [29,30].

## 4. Conclusions

The epoxy resin was modified by the addition of three types of CSR nanoparticles of different contents. On the one hand, the addition of all of the additives containing the soft CSR nanoparticles resulted in a minor decrease in the density, and a substantial reduction in the elastic modulus and tensile strength of the epoxy resin. The Hansen model was applied to describe the elastic modulus of the epoxy having a certain fraction of the CSR nanoparticles and pores, and a good agreement with the experimental results was found at the high CSR contents.

On the other hand, it was testified that the fracture toughness of the epoxy was significantly improved by the addition of all of the investigated types of CSR. The optimum CSR nanoparticle content was found to be 4 wt.% for all of the CSR nanoparticle types, thereby resulting in the improvement of the fracture toughness of the epoxy by 60–108%. No considerable distinction in the fracture toughness among the additives was detected, thereby proving that the small (100 nm) and large (300 nm) CSR nanoparticles were equally efficient.

Moreover, the effect of all of the additives containing the CSR nanoparticles was substantial, leading to the improvement in the interlaminar fracture toughness of the CFRP by 32–53%. Although, the toughening effect of the CSR nanoparticles in the epoxy was two times higher than it was in the epoxy-based CFRP composite laminates.

Additionally, a gradual increase of the glass transition temperature was obtained for the epoxy that was filled with all of the additives containing CSR nanoparticles, which could be attributed to the high crosslink density and toughening effect of rubber modifiers, thereby testifying to their dissolution in the epoxy continuous phase.

The possible combination of rigid and soft particles could be a compromise to simultaneously improve both the tensile properties and the fracture toughness, which cannot be achieved by the single-phase particles independently.

## Figures and Tables

**Figure 1 materials-15-07502-f001:**
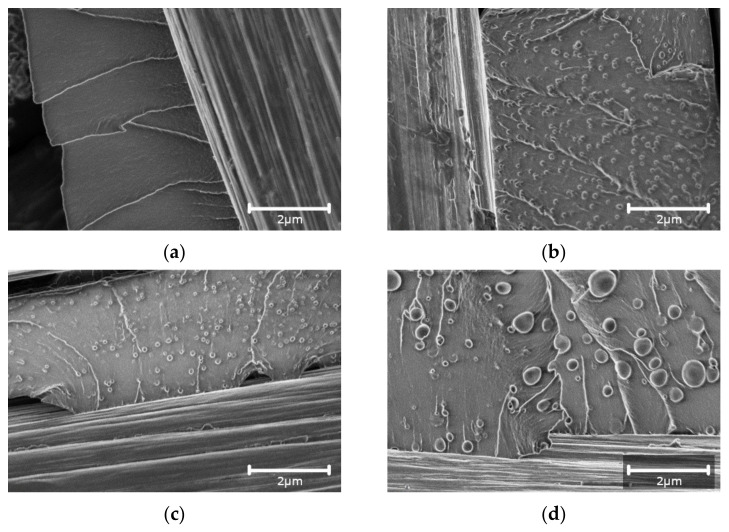
SEM images of fracture surface for the CFRP impregnated with the neat epoxy (**a**) and epoxy/CSR particles (4 wt.%) for the additives: ACE MX-125 (**b**), ACE MX-156 (**c**), and ACE MX-960 (**d**) (scale—2 µm, magnification—×25000).

**Figure 2 materials-15-07502-f002:**
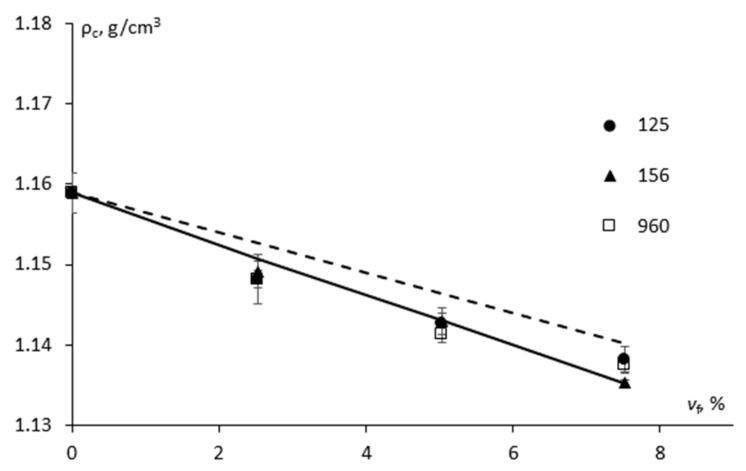
The density of the epoxy modified with different additives containing CSR nanoparticles (indicated on the graph) as a function of filler volume fraction (symbols—experimental data, dashed and solid lines—estimation by Equations (8) and (10), respectively.

**Figure 3 materials-15-07502-f003:**
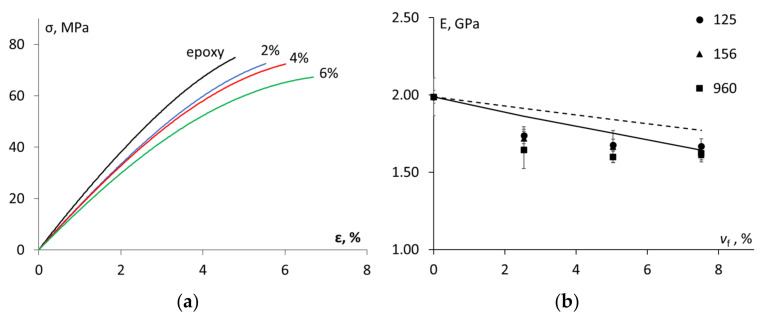
Stress–strain curves for the epoxy and epoxy filled with ACE MX-156 at different filler fractions indicated on the graph (**a**) and elastic modulus vs. filler volume fraction (dots -experimental results for different CSR nanoparticles, dashed and solid lines—evaluation by Equation (14) and by Equation (12), respectively (**b**).

**Figure 4 materials-15-07502-f004:**
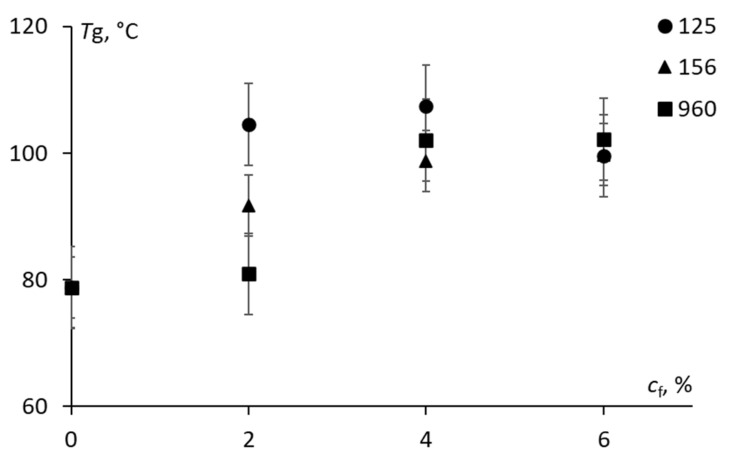
The glass transition temperature of the epoxy modified with different additives containing CSR nanoparticles (indicated on the graph) as a function of filler weight fraction.

**Figure 5 materials-15-07502-f005:**
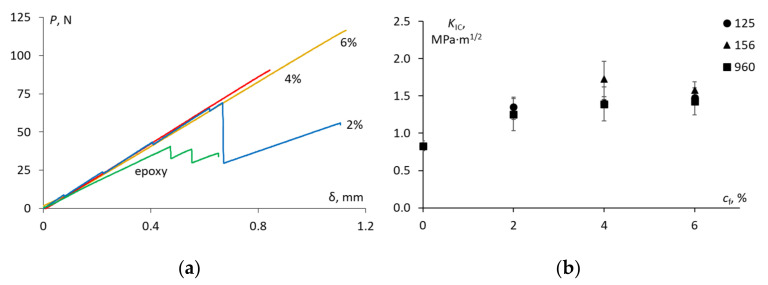
Load–displacement curves for TDCB tests of the epoxy and epoxy filled with ACE MX-960 at different filler weight fractions indicated on the graph (**a**) and fracture toughness for the epoxy and epoxy modified with ACE MX-125, 156 and 960 at different weight filler fractions (indicated on the graph) (**b**).

**Figure 6 materials-15-07502-f006:**
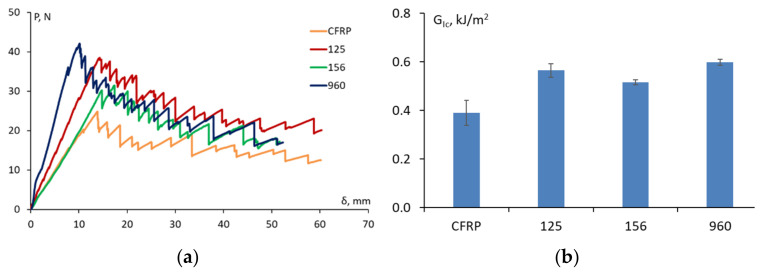
Typical load-crack opening displacement curves for CFRP and CFRP modified with ACE MX (indicated on the graph) at 4 wt.% (**a**) and interlaminar fracture toughness evaluated by Equation (6) for different materials studied (indicated on the graph) (**b**).

**Table 1 materials-15-07502-t001:** CSR types dispersed in the epoxy [12].

Additive Name	Core Material	CSR Size, nm
ACE MX-125	Styrene butadiene	100
ACE MX-156	Polybutadiene	100
ACE MX-960	Siloxane	300

## Data Availability

Not applicable.

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
