# Peer review of "Effect of Core–Shell Rubber Nanoparticles on the Mechanical Properties of Epoxy and Epoxy-Based CFRP"

_materials, 2022, doi:10.3390/ma15217502_

Round 1

Reviewer 1 Report

The manuscript can be accepted for publication. 

A few comments are given. 

1. CFRP- At least once the expanded form can be given. 

2. Figure 2: Why the density is decreasing?

3. Thermomechanical test is detailed in the experimental section. Also, conclusions include a statement based on that. However, it's data is not explained well in in the manuscript. Better to add a graph or table based on the TMA data.

4. Is it possible to add FTIR data of the raw materials and the laminates along with these results?

Author Response

Please see the attachment."

Reviewer 2 Report

General remarks

Subject of the article is interesting and the proposed title is compatible with the content. 

The introduction should contain more information about the applicability of the proposed solutions.

Specific remarks  

Line 12 - Please expand the CFRP abbreviation that it is carbon fiber reinforced polymer

Lines 86-87 - The description shows that five dog-bone samples and five tapered double cantilever beam (TDCB) samples were prepared. The next sentence (line 89) contains information about double cantilever beam (DCB) samples. So how many samples were prepared in total?
